# Threefold coordinated germanium in a GeO$_2$ melt

Songming Wan [1,2,3] ✉, Shujie Zhang[1], Bin Li[1], Xue Zhang[1], Xiaoye Gong[2] & Jinglin You[2]

The local structure around germanium is a fundamental issue in material science and geochemistry. In the prevailing viewpoint, germanium in GeO$_2$ melt is coordinated by at least four oxygen atoms. However, the viewpoint has been debated for decades due to several unexplained bands present in the GeO$_2$ melt Raman spectra. Using in situ Raman spectroscopy and density functional theory (DFT) computation, we have found a [GeOØ$_2$]$_n$ (Ø = bridging oxygen) chain structure in a GeO$_2$ melt. In this structure, the germanium atom is coordinated by three oxygen atoms and interacts weakly with two neighbouring non-bridging oxygen atoms. The bonding nature of the chain has been analyzed on the basis of the computational electronic structure. The results may settle down the longstanding debate on the GeO$_2$ melt structure and modify our view on germanate chemistry.

## Results and discussion

The GeO$_2$ melt is the archetype of various germanate melts. Its structure, particularly the local coordination environment of germanium, is of technological and geochemical importance and has been investigated extensively using various experimental methods or theoretical simulations[1–4]. It is widely accepted that germanium in the GeO$_2$ melt is coordinated by four oxygen atoms at ambient conditions and by six oxygen atoms at extreme conditions[5]. However, the viewpoint still remains controversial. A long-standing puzzle arises from the Raman spectrum. The typical Raman spectrum of the GeO$_2$ melt is dominated by a broad peak centered at 420 cm$^{-1}$, with two shoulders (X$_1$ and X$_2$) present around 340 and 520 cm$^{-1}$. About the structural origins of the two shoulders, the debates have continued for decades[6–10]. In this work, we investigate the Raman spectra and the bonding nature of a GeO$_2$ melt by density functional theory (DFT) computation, focusing, in particular, on the structural origins of the two mysterious shoulders, and demonstrate threefold coordinated germanium occurring in the melt.

### Results and discussion

#### Experimental Raman spectra

A polycrystalline GeO$_2$ sample was heated in a Linkam TS1500 microscopic furnace. Its Raman spectra were recorded in situ from room temperature through its melting temperature (1388 K) on a Horiba Jobin Yvon LabRAM HR Evolution Raman spectrometer (for details, see Methods). The spectra, before melting, exhibit the typical features of quartz-type GeO$_2$[11]. After melting, some new Raman bands, centered at 340 (shoulder X$_1$), 520 (shoulder X$_2$), 735, and 805 cm$^{-1}$, are present and systematically increase in intensity with temperature (Fig. 1).

#### Structural transformation on melting

Recent studies have pointed out that heat-induced structural transitions of solids, including melting, often involve decreases in the coordination number of composed atoms[12–15]. For example, GeO$_2$ undergoes the structural transition from the rutile- to quartz-type when heated at 1281 K (Fig. 2). Accompanying the transition, the coordination number of germanium decreases from six to four[16]. The decrease is probably associated with oxygen atom vibration (Fig. 2). At low temperatures, GeO$_2$ is formed by octahedral [GeØ$_6$] (rutile-type GeO$_2$, Ø = bridging oxygen), and each germanium atom is coordinated by six oxygen atoms with small vibrational amplitudes. At 1281 K, the vibration of the oxygen atoms is enhanced, the space available around the germanium atom cannot accommodate six oxygen atoms, and thus octahedral [GeØ$_6$] has to transform to tetrahedral [GeØ$_4$] (the motif of quartz-type GeO$_2$). For the same reason, the fourfold coordinated germanium in quartz-type GeO$_2$ will convert to lower-coordinated germanium when heated at a higher temperature, such as at the GeO$_2$ melting point (1388 K).

---

[1]Hefei Institutes of Physical Science, Chinese Academy of Sciences, Hefei 230031, China. [2]State Key Laboratory of Advanced Special Steel, Shanghai University, Shanghai 200444, China. [3]Advanced Laser Technology Laboratory of Anhui Province, Hefei 230037, China. ✉e-mail: smwan@aiofm.ac.cn

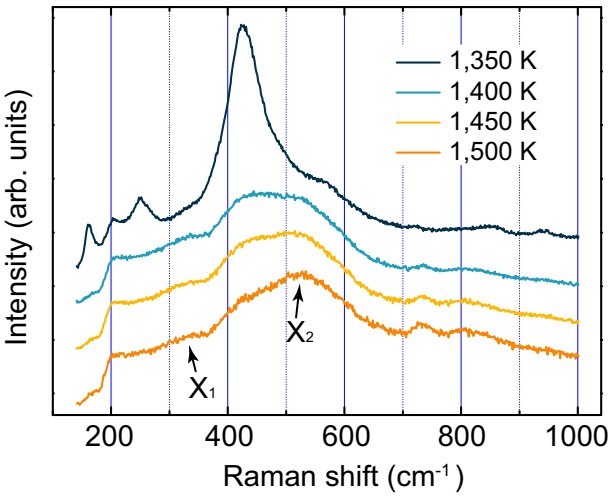

**Fig. 1 | Raman spectra of GeO₂ before and after melting.** GeO₂ melts at around 1388 K. X₁ and X₂ denote two characteristic Raman bands of the GeO₂ melt. Source data are provided as a Source Data file.

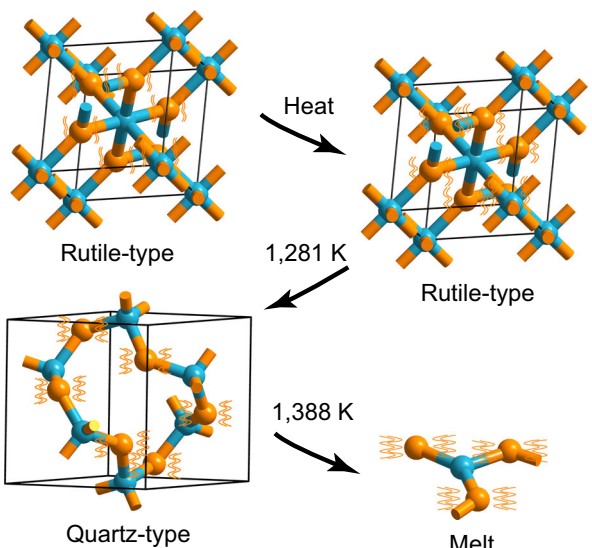

**Fig. 2 | Structural transformations of GeO₂ during heating.** Blue and orange spheres represent germanium and oxygen atoms, respectively. The orange wavy lines represent the vibration of oxygen atoms; their amplitude increases with temperature. Rutile-type GeO₂ crystallizes in the $P4_2/mnm$ space group (No. 136) with two GeO₂ formulae in a tetragonal unit cell ($a = 4.4066$ Å and $c = 2.8619$ Å), and features a continuous $[GeØ_6]_n$ network[28]. Quartz-type GeO₂ belongs to the $P3_221$ space group (No. 154) with three GeO₂ formulae in a trigonal unit cell ($a = 4.9870$ Å and $c = 5.6520$ Å), and features a continuous $[GeØ_4]_n$ network[29]. Rutile-type GeO₂ transforms to the quartz-type at 1281 K and then melts at 1388 K.

Figure 3 presents a possible structural transformation of GeO₂ in the melting process. After melting, some Ge−Ø bonds in quartz-type GeO₂ are broken (Fig. 3b), which yields the Ge−O bonds and threefold coordinated germanium atoms and finally the $[GeOØ_2]_n$ chain (Fig. 3c). Thus, the GeO₂ melt probably consists of the $[GeOØ_2]_n$ chain and the $[GeØ_4]_n$ network.

## Structural model and computational Raman spectrum

DFT computation was used to study the structural features of the $[GeOØ_2]_n$ chain and then to simulate its Raman spectrum (see Methods for computation details). Two $[GeOØ_2]_n$ chains with four GeO₂ formulae were placed into a three-dimensional, periodic, orthogonal unit

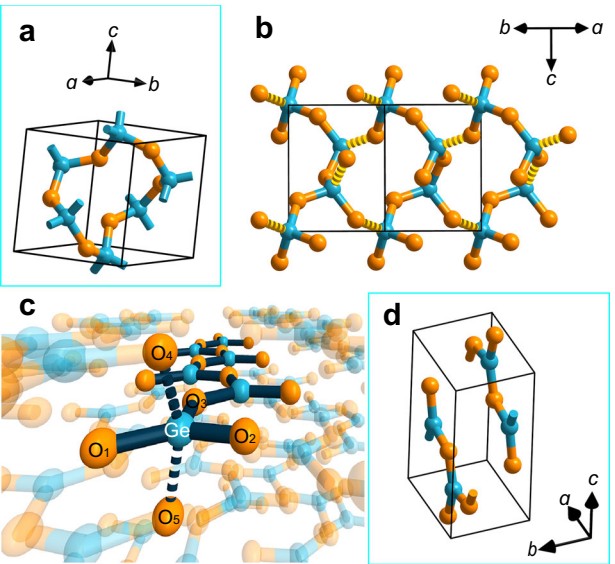

**Fig. 3 | GeO₂ structures before and after melting.** Blue and orange spheres represent germanium and oxygen atoms, respectively. **a** Quartz-type GeO₂ (before melting). **b** Structural change occurring in the melting process. The fragmented bonds are broken in this process; as a result, fourfold coordinated germanium (in the form of $[GeØ_4]$) converts to threefold coordinated germanium (in the form of $[GeOØ_2]$). **c** A $[GeOØ_2]_n$ chain formed after melting. **d** Structural model of the $[GeOØ_2]_n$ chain.

cell, with fixed parameters $a = 4.85$ Å, $b = 3.82$ Å, and $c = 8.15$ Å, to construct the structural model. The model has $D_{2h}^{16}$ symmetry (space group *Pnma*). The reliability of the computational method was confirmed by comparing the computational Raman spectrum of quartz-type GeO₂ to the experimental one (see Methods for details). The optimized structure of the $[GeOØ_2]_n$ chain is shown in Fig. 3d (see Supplementary Fig. 1 and Supplementary Table 1 for more details). In this structure, each germanium atom is surrounded by five oxygen atoms (Fig. 3c); three of them are from the same chain with short Ge−O distances ($d_{Ge−O1} = 1.84$ Å, $d_{Ge−O2} = 1.78$ Å, and $d_{Ge−O3} = 1.77$ Å) and two from two neighboring chains with long Ge−O distances ($d_{Ge−O4} = d_{Ge−O5} = 1.97$ Å). 1.97 Å is greater than the length of a normal Ge−O single bond (usually no more than 1.92 Å[17]), indicating that the germanium atom is threefold coordinated.

The reliability of the chain structure was evaluated by DFT computation. The computational total energy of the chain model is −3868.22 eV; accordingly, each $[GeOØ_2]$ motif has an energy of −967.06 eV. The energy is slightly higher than that of the $[GeØ_4]$ motif in quartz-type GeO₂ (−967.09 eV), revealing that $[GeOØ_2]$ is a metastable structure at low temperature and can coexist with $[GeØ_4]$ at high temperature.

The computational spectra of the $[GeOØ_2]_n$ chain and the $[GeØ_4]_n$ network (quartz-type GeO₂), along with the GeO₂ melt experimental spectrum (recorded at 1500 K), are shown in Fig. 4a. All of the experimental bands are produced in the computational spectra (see Supplementary Tables 2 and 3 for more details). The computational bands are in agreement with the experimental ones, not only in frequency but also in intensity, which confirms that the $[GeOØ_2]_n$ chain coexists with the $[GeØ_4]_n$ network in the GeO₂ melt. To the best of our knowledge, the $[GeOØ_2]_n$ chain is the first example of a species with threefold coordinated germanium.

According to the DFT computational results, both the shoulders X₁ and X₂ are associated with the $[GeOØ_2]_n$ chain and arise from the vibration of the bridging oxygen along the Ge−O−Ge angle bisection (Fig. 4b, c). Unlike X₂, X₁ involves the distinctive stretching vibration of the germanium atom along the Ge−O bond. Atomic vibrations for

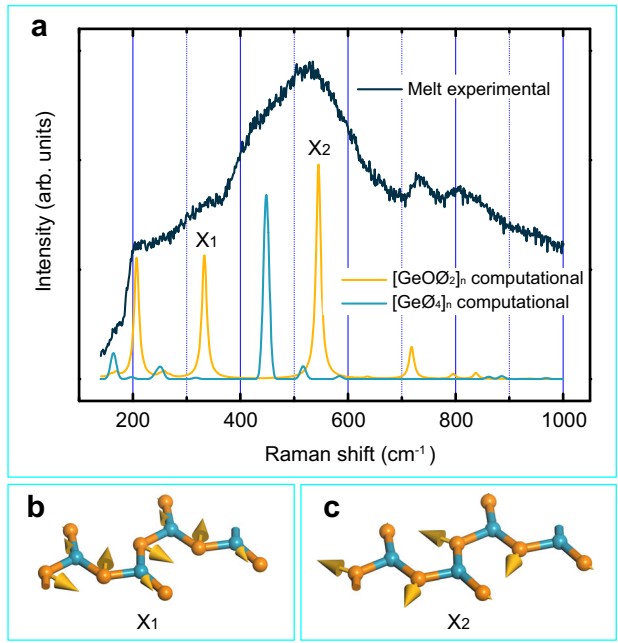

**Fig. 4 | Computational results for the GeO₂ melt.** Blue and orange spheres represent germanium and oxygen atoms, respectively. **a** Computational Raman spectra of the $[GeO\text{Ø}_2]_n$ chain and the $[Ge\text{Ø}_4]_n$ network, together with the experimental spectrum of the GeO₂ melt (recorded at 1500 K). **b** Atomic vibrations for shoulder $X_1$. **c** Atomic vibrations for shoulder $X_2$. Source data are provided as a Source Data file.

other strong Raman bands of the $[GeO\text{Ø}_2]_n$ chain are available in Supplementary Fig. 2. It is noteworthy that the Raman bands due to the Ge–O stretching vibrations are located at around 715 cm⁻¹ (Supplementary Figs. 2b and 3b, c), significantly lower than the Ge–O stretching frequency of the $[GeO_2\text{Ø}_2^{2-}]_n$ chain (located at around 815 cm⁻¹)[18,19], which reflects that the Ge–O bonding in $[GeO\text{Ø}_2]_n$ is weaker than that in $[GeO_2\text{Ø}_2^{2-}]_n$.

### Electronic structures

Knowledge of the electronic structure is critical for understanding the nature of chemical bonding in the $[GeO\text{Ø}_2]_n$ chain. Here, only the valence electrons (Ge-$4s^2 4p^2$ and O-$2s^2 2p^4$) are considered for the DFT computation of the electronic structure. The total and projected densities of states (DOSs) of $[GeO\text{Ø}_2]_n$ are shown in Fig. 5a. All the bonding orbitals of the germanium atom are composed of the Ge-$4s$ and Ge-$4p$ orbitals and display some hybrid character, especially the orbitals in the energy range from −19.5 to −15.5 eV. The bonding orbitals from −9.0 to −6.0 eV and from −6.0 to 0.0 eV mainly derive from the Ge-$4s$ and Ge-$4p$ orbitals, respectively, apart from the O-$2p$ orbitals. In comparison with quartz-type GeO₂ (Fig. 5b), the $[GeO\text{Ø}_2]_n$ chain has more high-energy orbitals originating from the Ge-$4p$ and O-$2p$ atomic orbitals, which implies that the $[GeO\text{Ø}_2]_n$ chain has more weak- or non-bonding $p$ orbitals. The electron localization function (ELF) is a measure of the probability of finding an electron pair in a space region and is an intuitive tool to identify the character of a chemical bond[20]. A valence ELF map of $[GeO\text{Ø}_2]_n$ is displayed in Fig. 5c. Most electron pairs are localized in the regions between Ge and O, which reveals the covalent nature of the Ge–O/Ge–Ø bonds. The population of the electron pairs between Ge and O in $[GeO\text{Ø}_2]_n$ resembles that in $[Ge\text{Ø}_4]_n$ (Fig. 5d), further confirming that the Ge–O/ Ge–Ø bonds have a similar character in the two structures.

The trigonal planar geometry of the $[GeO\text{Ø}_2]$ motif suggests that the germanium atom in the $[GeO\text{Ø}_2]_n$ chain is the $sp^2$ hybridization. In the hybridization, the Ge-$4s$ orbital mixes with two Ge-$4p$ orbitals to form three Ge-$sp^2$ orbitals of equal energy. At the same time, a Ge-$4s$

electron is excited to the empty Ge-$4p$ orbital and results in the $4s^1 4p_x^1 4p_y^1 4p_z^1$ configuration (Fig. 6a). Each Ge-$sp^2$ orbital electron pairs with an O-$2p$ electron, forming three σ bonds (Ge–O/Ge–Ø bonds). The remaining perpendicular $4p_z^1$ orbital is not involved in the bonding (Fig. 6b) but can interact weakly with two neighboring O-$2p$ orbitals.

Four representative bonding orbitals of $[GeO\text{Ø}_2]_n$ are shown in Fig. 7a. Except for the orbitals near the Fermi energy level, all of the bonding orbitals have the σ bond character. Since the O-$2p$ orbital is significantly different in size from the Ge-$4p$ orbital, the expected π bond, formed by laterally overlapping the Ge-$4p$ orbital with the three O-$2p$ orbitals, is not found. The $[GeO\text{Ø}_2]_n$ chain has similar bonding features to quartz-type GeO₂ except for the orbitals in the energy range from −8.6 to −6.2 eV. In the energy range, each germanium atom interacts with two non-bridging O atoms in the adjacent $[GeO\text{Ø}_2]_n$ chains (Fig. 7a). Nonetheless, the interaction is weaker than the typical Ge–O covalent bonding (Fig. 7b) and thus easily appears and disappears in the GeO₂ melt. We refer to the unique interaction as the fluxional bonding. The bonding can interpret the fluidity of the melt.

More of the bonding characteristics of the $[GeO\text{Ø}_2]_n$ chain are revealed by the Hirshfeld charge population analysis[21]. The Hirshfeld charges of the germanium, non-bridging oxygen, and bridging oxygen atoms in the chain are +0.54 e, −0.25 e, and −0.29 e, respectively, indicating that all the Ge–O/Ge–Ø bonds are polar. According to the ELF analyses, the electron pairs around the Ge–O bonds in the $[GeO\text{Ø}_2]_n$ chain are less than that in the $[GeO_2\text{Ø}_2^{2-}]_n$ chain (see Supplementary Fig. 4 for more details). The result is consistent with the Ge–O bond length data of the two chains (1.84 Å in $[GeO\text{Ø}_2]_n$ and 1.733 Å in $[GeO_2\text{Ø}_2^{2-}]_n$) and further supports our inference that the Ge–O bonding in $[GeO\text{Ø}_2]_n$ is weaker than that in $[GeO_2\text{Ø}_2^{2-}]_n$.

The threefold coordinated germanium, as well as the $[GeO\text{Ø}_2]_n$ chain, provides an insight into the structure of germanate melts, which is helpful for better understanding the melt behaviors in various technological processes such as crystal growth and glass production. Besides, the GeO₂ melt is widely considered a chemical and structural analog of the SiO₂ melt, the main constituent of magmas[1,3,22]. Hence the present GeO₂ structure may have important implications for exploring the geochemical evolution occurring in Earth's interior.

## Methods

### Raman spectroscopy

A GeO₂ polycrystalline sample (rutile-type, 99.999%, Sinopharm Chemical Reagent) was placed into a platinum crucible which was heated in a Linkam TS1500 microscopic furnace. The Raman spectra of the sample were recorded on a Horiba Jobin Yvon LabRAM HR Evolution Raman spectrometer. The excitation source was the 355 nm line delivered by a Q-switched THG Nd:YAG pulsed laser with a power of about 10 mW. Raman scattering light was collected using an optical confocal system in a backscattering configuration. The Raman scattering light was analyzed with a single grating monochromator and detected with a CCD camera. The spectral resolution was about 1.3 cm⁻¹. Prior to Raman measurements, the spectrometer was calibrated using a silicon wafer.

### DFT computations

DFT computations within the plane-wave/pseudopotential scheme were performed using the Cambridge Sequential Total Energy Package (CASTEP)[23]. The generalized gradient approximation (GGA) in the Wu–Cohen (WC) parametrization was adopted to treat the exchange and correlation effects. The use of the GGA-WC functional can significantly improve the computational accuracy relative to the most popular GGA-PBE (Perdew–Burke–Ernzerhof) functional[24]. Norm-conserving pseudopotentials were employed to describe the core–electron interactions. The valence electron configurations of germanium and oxygen were $4s^2 4p^2$ and $2s^2 2p^4$, respectively. An energy cutoff of 750 eV

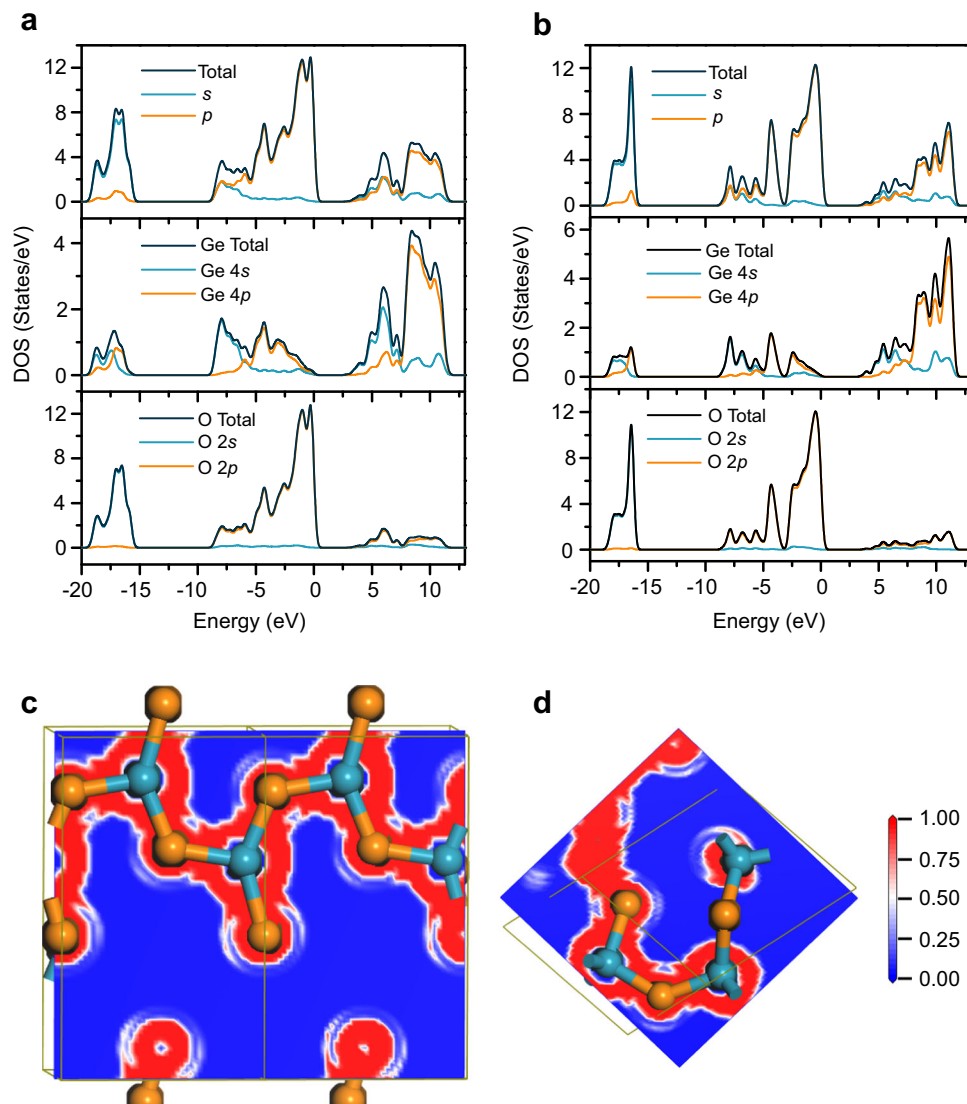

**Fig. 5 | Electronic structures of the two structures in the GeO₂ melt.** Blue and orange spheres represent germanium and oxygen atoms, respectively. **a** Total and projected DOSs for the [GeOØ₂]ₙ chain. **b** Total and projected DOSs for the [GeØ₄]ₙ network. **c** A valence ELF map for the [GeOØ₂]ₙ chain. **d** A valence ELF map for the [GeØ₄]ₙ network. The maps are along the planes on which the Ge−O/Ge−Ø bonds lie. Source data are provided as a Source Data file.

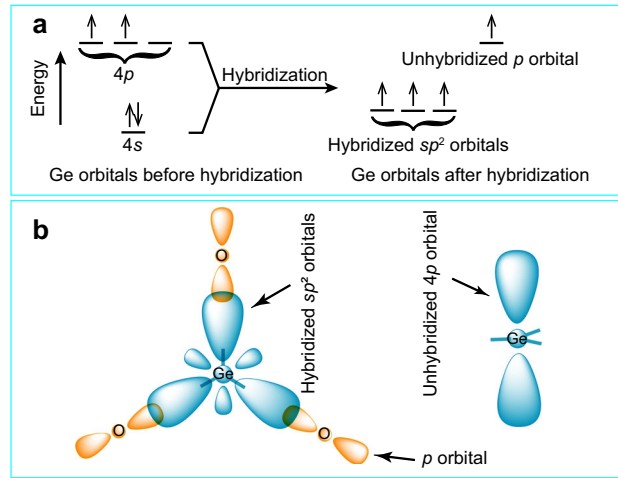

**Fig. 6 | Hybridization and bonding in the [GeOØ₂] motif. a** Hybridization scheme for the germanium atom. **b** Interactions between Ge-$sp^2$ and O-2$p$ orbitals (left), and the unhybridized Ge-4$p$ orbital (right).

was used in all computations. The Brillouin zone integrations were performed over a $3 \times 3 \times 2$ Monkhorst Pack grid for quartz-type GeO₂ and over a $3 \times 4 \times 2$ grid for the [GeOØ₂]ₙ chain. The two structural models were optimized until the total energy change, maximum displacement, maximum force, and maximum stress were less than $10^{-6}$ eV/atom, 0.001 Å, 0.03 eV/Å, and 0.05 GPa, respectively. For quartz-type GeO₂, the cell parameters, as well as atomic positions, were optimized. For the [GeOØ₂]ₙ chain, all the atomic positions were optimized in a unit cell with fixed parameters $a = 4.85$ Å, $b = 3.82$ Å and $c = 8.15$ Å. After structure optimizations, the DOSs, ELFs, Hirshfeld charge populations, and bonding orbitals were computed with CASTEP for the two structural models.

CASTEP uses density functional perturbation theory (DFPT, also known as the linear response method) to compute the Raman spectra (modes, frequencies, and intensities at the $\Gamma$ point) of quartz-type GeO₂ and the [GeOØ₂]ₙ chain[25]. By constructing the Hessian matrix, Raman vibrational information was obtained. The eigenvectors of the matrix are the Raman modes; the square roots of the eigenvalues are the Raman frequencies. The intensity of each vibrational mode was computed from the derivative of the dielectric polarizability tensor with

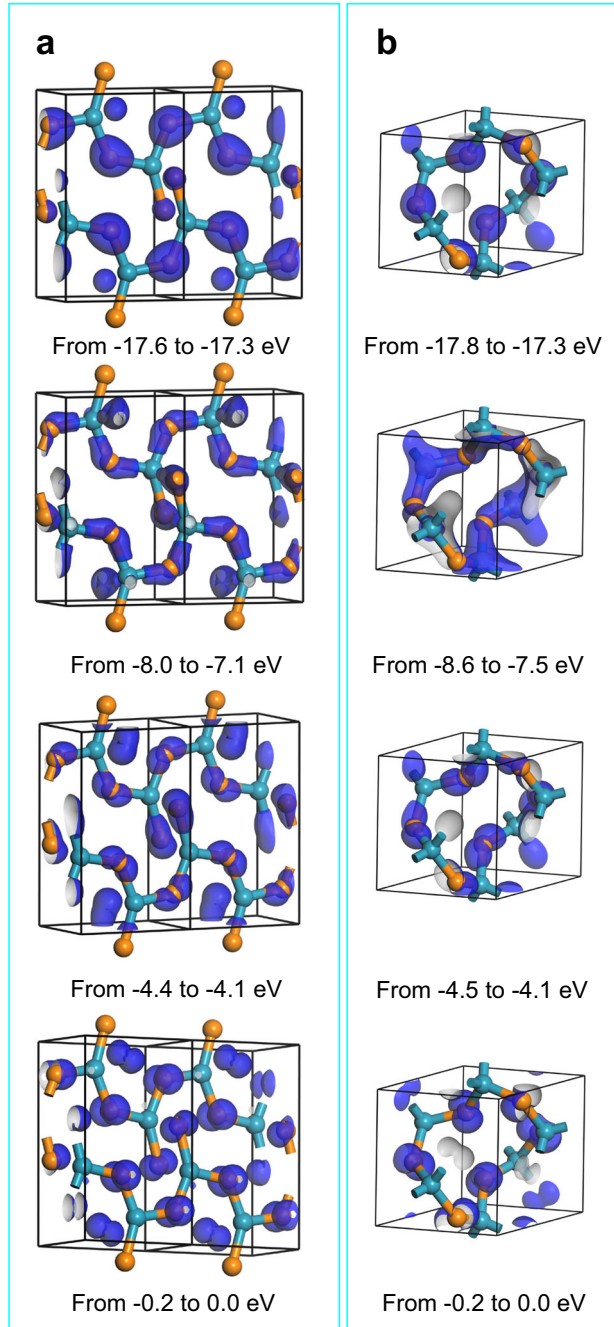

**Fig. 7 | Representative orbitals in different energy ranges for the two structures in the GeO₂ melt.** Blue and orange spheres represent germanium and oxygen atoms, respectively. **a** The [GeOØ₂]ₙ chain. **b** The [GeØ₄]ₙ network.

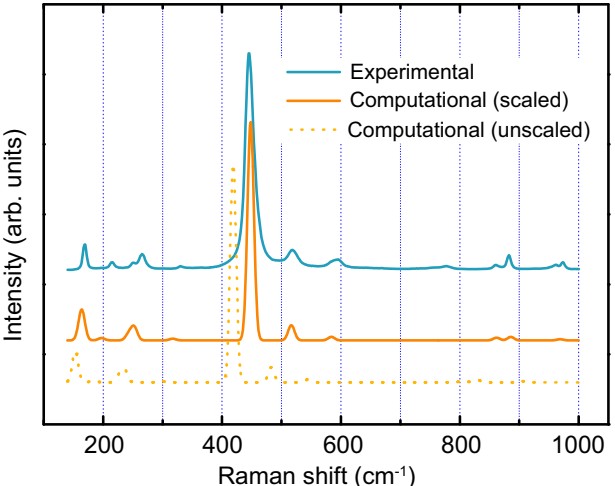

**Fig. 8 | Computational and experimental Raman spectra of quartz-type GeO₂.** The computational frequencies are scaled by 1.070. Source data are provided as a Source Data file.

quartz-type $GeO_2$ is shown in Fig. 8, which is in good agreement with the experimental spectrum both in frequency and intensity.

### Reporting summary
Further information on research design is available in the Nature Portfolio Reporting Summary linked to this article.

## Data availability
The data supporting the findings of this study are available within the article and its Supplementary Information. Additional data are available from the corresponding author upon request. Source data are provided in this paper.

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

## Acknowledgements

This work was financially supported by the National Natural Science Foundation of China (grant Nos. 51372246 (S.W.) and 21773152 (J.Y.)), the HFIPS Director's Fund (grant No. YZJJ202301-TS (S.W.)), Open Project of Advanced Laser Technology Laboratory of Anhui Province (grant No. AHL2022ZR04 (S.W.)), and Open Project of Key Laboratory of Functional Crystals and Laser Technology, TIPC, Chinese Academy of Sciences (grant No. FCLT 202001 (S.W.)). The DFT computations were performed in part at the Center for Computational Science, CASHFIPS.

## Author contributions

S.W. initiated the work. S.Z. and X.G. conducted the Raman experiments. S.W., B.L., and X.Z. performed the DFT computations. S.W. and J.Y. supervised the work. All authors participated in the discussion.

## Competing interests

The authors declare no competing interests.
