## [Peer Review File · Nature Communications]

REVIEWERS' COMMENTS (first round)

Reviewer #1 (Remarks to the Author):

I have examined the article "Threefold coordinated germanium in a GeO₂ melt" by Songming Wan et al. This article represents a joint experimental and theoretical investigation on a topic relevant to various research fields, i.e. chemistry, solid state physics and geology. Overall, this is a nice paper, well written and organized. The quality of artworks is adequate, references cover the subject properly, the methods adopted by the authors are appropriate.

The subject investigated by the authors falls in a more general topic, i.e. the coordination and the chemistry of group IV elements. In the structure proposed by the authors, germanium atoms are three-fold coordinated and arrange in planar "sp²" like arrays. In carbon- and in silicon-based compounds such arrangement implies the formation of electronic states with Pi character. Is this the case of three-fold coordinated GeO₂?

The above question leads to my first general observation on the article. The authors did not perform the analysis of the electronic structure of the three-fold coordinated structure. Its comparison with four-fold germanium dioxide would be of great help in establishing the nature of planar GeO₂ arrays. I also note that in "three-fold" germanium the metal atoms are bonded to three oxygens at the typical Ge–O internuclear separation, but they face also two O at a 15% elongated distance. What is the effect of such coordination on the nature of bonds? How does the electronic band structure react to such structural arrangement? I encourage the authors to answer these questions.

I also have a set of technical questions that still deserve an answer before publication is considered:

What is the technical meaning of optimizing the "structural models"? Does it correspond to the optimization of the cell parameters as well as nuclear positions? Is the three-fold coordinated GeO₂ structure a minimum in the phase space of the possible structures? Did the authors compare total energy of four-fold and three-fold GeO₂? This last issue might help in evaluating the reliability of the model proposed in this article.

In conclusion, I believe that the topic raised by the authors is relevant and of potential interest for a wide community of researchers. Results are interesting, and they are presented clearly. However, there are several major issues that should be answered before considering publication.

Reviewer #2 (Remarks to the Author):

The manuscript presents the results of Raman spectroscopy and density functional theory (DFT) computation of GeO₂ melts. This is an interesting attempt, which could lead to a better understanding of the GeO₂ structure as a function of temperature. However, some points are unclear:

1. According to results of the study the main structural unite of the melt is assumed to be [GeO₃] motif. Figure 2D contains the picture of them. As I can understand there are three Ge–O bonds. Are they identical?
2. How many valence electrons do germanium and oxygen have? As I know there were three sigma bonds and one pi bond around Ge atom in GeO₂ at room temperature. Can you explain the structure of [GeO₃] motif in melt according to this information?

3. Two bands near 870 and 970 are TO/LO asymmetric stretching of bridging oxygens in GeO₂ glasses (Galeener, 1980; Henderson and Fleet, 1991; Micoulaut et al., 2006; Verweij and Buster, 1979). The lines positions in the Raman spectrum demonstrate a shift of frequencies to the low-frequency region with temperature – this is a well-known fact (Koroleva et al., 2013; Koroleva and Osipov, 2020; Osipov and Osipova, 2009). According to Figure 1 two bands about 750 and 810 are the result of such a shift. Check this by tracking the changes in the spectrum with a temperature from 298.15 to 1500 K.
4. The structural motif of Li₂GeO₃ is the [GeO₂Ø₂]²⁻ containing two negative charges, not [GeO₂Ø₂]. What's why spectra of methagermanate crystals and glasses are characterized by the band in the region about 800 cm⁻¹ corresponding to vibrations of Ge–O⁻ bonds? How can you explain appearance of bands in the region of vibrations of carrying negative charges non-bridging oxygens? Are there negative charges in the GeO₂ melt?

References:

1. Galeener, F. L. 1980. The Raman spectra of defects in neutron bombarded and Ge-rich vitreous GeO₂. *J. Non-Cryst. Solids*. 40: 527–533.
2. Henderson, G. S. and M. E. Fleet. 1991. The structure of alkali germanate and silicate glasses by Raman spectroscopy. *Trans. Am. Crystallogr. Assoc.* 27: 269–278.
3. Koroleva, O. N., V. N. Anfilogov, Shatskiy A. and L. K. D. 2013. Structure of Na₂O–SiO₂ melts as a function of composition: in situ Raman spectroscopic study. *J Non-Crystalline Solids*. 375.
4. Koroleva, O. N., and A. A. Osipov. 2020. In situ Raman spectroscopy of K₂O–GeO₂ melts. *Journal of Non-Crystalline Solids*. 531: 119850.
5. Micoulaut, M. L. Cormier, and G. S. Henderson. 2006. The structure of amorphous, crystalline and liquid GeO₂. *Journal of Physics: Condensed Matter*. 18: R753.
6. Osipov, A. A. and L. M. Osipova. 2009. Structure of glasses and melts in the Na₂O–B₂O₃ system from high-temperature Raman spectroscopic data: I. Influence of temperature on the local structure of glasses and melts. *Glass Physics and Chemistry*. 35: 121–131.
7. Verweij, H. and J. H. J. M. Buster. 1979. The structure of lithium, sodium and potassium germanate glasses, studied by Raman scattering. *J. Non-Cryst. Solids*. 34: 81–99

Reviewer #3 (Remarks to the Author):

This is an interesting computational and experimental paper on the Raman spectra of GeO₂ melts identifying a new type of structure involving a 3-fold coordinated Ge in the melt. The work overall is quite nice, but the presentation needs work. Too much of the critical information is in the SI and a reader should not have to have the SI open to read the paper. In this case, one cannot read the paper without the SI present. In addition, the work would be hard to reproduce given the information in the paper.

Line 52. Reword vigorous atomic vibration as this does not make any sense. I would suggest thinking about using the term fluxional bonding as bonds are readily made and broken in the melt. The same is true for line 53 as I have no idea what six vibrant oxygen are. Vibrant is not properly used in this contest as it has a very different meaning.

In Figure 2, what is the source of the data for the pictures before and after melting? This needs to be given in the text or figure caption. Figure S2 shows what is going on much better than the current Figure 2

and should be added to it.

Please show a picture of the Li_2GeO_3 crystal in the text. I think that Figure S1 is critical to the paper and should be in the text.

Figure S3 is a critical benchmark and should be in the computational methods section in the text.

I really do not think that their bond distances in the melt are good to 3 decimal places in angstroms if these are from dynamics type simulations which they need to be at the higher temperature. They are likely to be average values and good to only 2 decimal places.

The authors note that there are allowed and forbidden IR and Raman bands in Table S2. This can only happen if symmetry is present. Is there symmetry in the chain? This needs to be discussed in more detail. Also, as they have the values, the Raman and IR intensities of all these transitions need to be given.

Where are the computed Raman and IR intensities and the associated frequencies for the melt and the pure crystal? These need to be given.

The high frequency bands are likely to be Ge–O stretches based on what I know of the bands in metal oxides. Does the number of Ge–O stretches correspond to the numbers of Ge–O bonds in the calculations? How many Ge and O are there in the chain calculations used for Table S2?

Details of the Raman calculations and how the Raman intensities were calculated and corrected need to be in the Supporting Information in much more detail to allow the results to be reproduced. I would think that the line widths seem to be quite narrow for comparison to the experiment in the melt.

I am not sure how relevant GeO_2 chemistry is to SiO_2 chemistry as stated in the paper as the Si–O bonds are stronger than the Ge–O bonds so will require higher temperatures. Do they expect to find the same structures in SiO_2 melts? As this is discussed in the conclusions, the authors need to comment on the relevance of this to the actual SiO_2 chemistry they want to mimic for modeling the Earth's core.

Overall, some nice work but the paper is too terse, and the work cannot readily be reproduced without more computational details. Figures from the SI need to be moved to the text.

Response to the comments from Reviewers (first round)

Thanks very much for the valuable comments on our manuscript entitled "Threefold coordinated germanium in a GeO₂ melt" (NCOMMS-22-43430). The Main Text and the Supplementary Information have been revised according to the comments. The point-by-point responses are as follows:

Reviewer #1 (Remarks to the Author):

I have examined the article "Threefold coordinated germanium in a GeO₂ melt" by Songming Wan et al. This article represents a joint experimental and theoretical investigation on a topic relevant to various research fields, i.e. chemistry, solid state physics and geology. Overall, this is a nice paper, well written and organized. The quality of artworks is adequate, references cover the subject properly, the methods adopted by the authors are appropriate.

1. The subject investigated by the authors falls in a more general topic, i.e. the coordination and the chemistry of group IV elements. In the structure proposed by the authors, germanium atoms are three-fold coordinated and arrange in planar "sp²" like arrays. In carbon- and in silicon-based compounds such arrangement implies the formation of electronic states with Pi character. Is this the case of three-fold coordinated GeO₂?

Fig. R1 | Total and projected densities of states for the [GeO_{0.2}]_n chain.

In order to answer this question, we studied the total and projected densities of states (TDOS and PDOS) and the bonding orbitals for the [GeO_{0.2}]_n chain. The results show that all the bonding orbitals of the Ge atom are composed of the Ge-4s and Ge-4p atomic orbitals and display some hybrid character, especially the orbitals in the energy range from -19.5 to -15.5 eV (Fig. R1). The bonding orbitals in the energy ranges from -9.0 to -6.0 eV and from -6.0 to 0.0 eV mainly derive from the Ge-4s and Ge-4p orbitals, respectively. Four representative bonding orbitals are shown in Fig. R2. All of the bonding orbitals, except those near the Fermi energy level, have the σ bond character. The expected π bond, formed by a Ge-4p orbital overlapping with O-2p orbitals side by side, is not found. This is probably due to the significant difference in size between the Ge-4p and the O-2p orbitals.

The relevant content, including Figs. R1 and R2, has been added into the main text (Lines 117 to 121, lines 129 to 132, Fig. 5a and Fig.6a).

Fig. R2 | Four representative bonding orbitals of the $[\text{GeO}_{0.25}]_n$ chain in different energy ranges. (a) From -17.3 to -17.6 eV, (b) from -8.0 to -7.1 eV, (c) from -4.4 to -4.1 eV and (d) from -0.2 to 0.0 eV. Olive and red spheres represent Ge and O atoms, respectively.

2. The above question leads to my first general observation on the article. The authors did not perform the analysis of the electronic structure of the three-fold coordinated structure. Its comparison with four-fold germanium dioxide would be of great help in establishing the nature of planar GeO_2 arrays. I also note that in "three-fold" germanium the metal atoms are bonded to three oxygens at the typical Ge-O internuclear separation, but they face also two O at a 15% elongated distance. What is the effect of such coordination on the nature of bonds? How does the electronic band structure react to such structural arrangement? I encourage the authors to answer these questions.

Fig. R3 | Total and projected densities of states for quartz-type GeO_2 .

Fig. R4 | Total and projected densities of states for quartz-type GeO_2 (top) and the $[\text{GeO}_{0.2}]_n$ chain (bottom).

In addition to the $[\text{GeO}_{0.2}]_n$ chain, we also studied the electronic structure of quartz-type GeO_2 which features a continuous GeO_4 3D network. The results are shown in Fig. R3. Compared with quartz-type GeO_2 , the $[\text{GeO}_{0.2}]_n$ chain has more high-energy bonding orbitals which originate from Ge-4p orbitals and O-2p orbitals (Figs. R1, R3 and R4). The result implies that the $[\text{GeO}_{0.2}]_n$ chain has more weak- or non-bonding orbitals.

Fig. R5 | Representative bonding orbitals of (a) the $[\text{GeO}_{0.2}]_n$ chain in the energy range from -8.0 to -7.1 eV, and (b) quartz-type GeO_2 in the energy range from -8.6 to -7.5 eV.

In the $[\text{GeO}_{0.2}]_n$ chain, each Ge atom is bonded by one non-bridging O atom and two bridging O atoms, and faces two O atoms with longer Ge–O distances. The DFT computation reveals that the Ge atom also interacts with the two long-distanced O atoms (Fig. R5a); however, the interaction is weaker than the typical Ge–O bonding (a typical Ge–O bonding orbital of quartz-type GeO_2 is shown in Fig. R5b). As suggested by Reviewer #3, the weak interaction is referred to as fluxional bonding which easily appears and disappears in the GeO_2 melt.

The relevant content, including Figs. R3 and R5, has been added into the main text (Lines 121 to 124, lines 132 to 137, Fig. 5b and Fig. 6).

3. What is the technical meaning of optimizing the "structural models"? Does it correspond to the optimization of the cell parameters as well as nuclear positions?

In our computations, the structural models of quartz-type GeO_2 and the $[\text{GeO}_{0.2}]_n$ chain had been optimized before computing their properties. For quartz-type GeO_2 , the cell parameters, as well as nuclear positions, were optimized. For the chain, only the nuclear positions were optimized.

The above content has been inserted into the Methods section (Lines 164 and 166).

4. Is the three-fold coordinated GeO₂ structure a minimum in the phase space of the possible structures?

To the best of our knowledge, the three-fold coordinated Ge is not the minimum in the phase space of GeO₂ structure. The GeO₂ molecule, isostructural with the CO₂ molecule, has been found by Bos et al. [Bos, A., Ogden, J. S. & Orgee, L. *J. Phys. Chem.* **78**, 1763–1769 (1974)]. In this structure, the Ge atom is coordinated by two oxygen atoms.

5. Did the authors compare total energy of four-fold and three-fold GeO₂? This last issue might help in evaluating the reliability of the model proposed in this article.

In accordance with the suggestion, we computed and obtained the total energies of the [GeO \emptyset_2]_n chain (–3868.2236 eV, four GeO₂ formulae in the unit cell) and quartz-type GeO₂ (–2901.2716 eV, three GeO₂ formulae in the unit cell). The energy per GeO \emptyset_2 (–967.056 eV) is slightly larger than that of Ge \emptyset_4 (–967.091 eV), which indicates that GeO \emptyset_2 is metastable at low temperature and can coexist with Ge \emptyset_4 at high temperature.

The above content has been added into the main text (Lines 88 to 91).

In conclusion, I believe that the topic raised by the authors is relevant and of potential interest for a wide community of researchers. Results are interesting, and they are presented clearly. However, there are several major issues that should be answered before considering publication.

Reviewer #2 (Remarks to the Author):

The manuscript presents the results of Raman spectroscopy and density functional theory (DFT) computation of GeO₂ melts. This is an interesting attempt, which could lead to a better understanding of the GeO₂ structure as a function of temperature. However, some points are unclear:

1. According to results of the study the main structural unit of the melt is assumed to be [GeO \emptyset_2] motif. Figure 2D contains the picture of them. As I can understand there are three Ge-O bonds. Are they identical?

The GeO \emptyset_2 motif has three covalent bonds. Two of them are the Ge– \emptyset bond, where \emptyset is the oxygen atom bonding to two Ge atoms; the remaining one is the Ge–O bond, where O is the oxygen atom bonding to only one Ge atom.

2. How many valence electrons do germanium and oxygen have? As I know there were three sigma bonds and one pi bond around Ge atom in GeO₂ at room temperature. Can you explain the structure of [GeO \emptyset_2] motif in melt according to this information?

The Ge atom has four valence electrons ($4s^2 4p^2$), and the O atom has six valence electrons ($2s^2 2p^4$). In order to learn about the bonding character of the GeO \emptyset_2 motif, we studied its electronic structure. The computational results are shown in Figs. R1 and R2. All of the Ge–O/Ge– \emptyset bonds have the σ bonding character. The expected π bond, formed by laterally overlapping of a Ge-4p orbital with O-2p orbitals, is not found, probably because the O-2p orbital is significantly different in size from the Ge-4p orbital.

The above explanation, as well as Figs. R1 and R2, has been added into the main text (Lines 129 to 132, Fig. 5a and Fig.6a).

3. Two bands near 870 and 970 are TO/LO asymmetric stretching of bridging oxygens in GeO₂ glasses

(Galeener, 1980; Henderson and Fleet, 1991; Micoulaut et al., 2006; Verweij and Buster, 1979). The lines positions in the Raman spectrum demonstrate a shift of frequencies to the low-frequency region with temperature—this is a well-known fact (Koroleva et al., 2013; Koroleva and Osipov, 2020; Osipov and Osipova, 2009). According to Figure 1 two bands about 750 and 810 are the result of such a shift. Check this by tracking the changes in the spectrum with a temperature from 298.15 to 1500 K.

Fig. R6 | Raman spectra of quartz-type GeO₂ recorded at 300 K and the GeO₂ melt recorded at 1300, 1400 and 1500 K.

Our computational results show that the bands near 870 and 970 cm⁻¹ arise from the vibrations of GeO₄ while the bands near 750 and 810 cm⁻¹ arise from the vibrations of GeO₂ (Fig. 4a in the main text).

In Fig. R6, we show the spectra of GeO₂ recorded at 1300, 1400, and 1500 K and quartz-type GeO₂ recorded at 300 K. At 1300 K, GeO₂ is quartz-type, which is made up of GeO₄ and thus the 860 and 940 cm⁻¹ bands in the 1300 K spectrum should originate from GeO₄. As temperature increases, the two bands will shift to lower frequencies and decrease in intensity. Although they are not intense enough to be observed above 1300 K, we believe that they are still located around 860 and 940 cm⁻¹ because, in a general way, the two bands cannot move to 750 and 810 cm⁻¹ in such a limited temperature range (1300–1500 K).

Fig. R7 | Raman spectra of GeO₂ recorded at 400, 800, 1200 K.

In order to track the spectral changes of GeO_2 , we cooled the GeO_2 melt from 1500 K to room temperature and collected its Raman spectra at different temperatures. Fig. R7 presents three typical spectra recorded below 1300 K. All of them have the characteristics of rutile-type GeO_2 , which is formed by GeO_6 , rather than $\text{GeO}_4/\text{GeO}_2$; therefore, here the structural changes of GeO_2 below 1300 K are not discussed.

4. The structural motif of Li_2GeO_3 is the $[\text{GeO}_2\text{O}_2]^{2-}$ containing two negative charges, not $[\text{GeO}_2\text{O}_2]$. What's why spectra of metagermanate crystals and glasses are characterized by the band in the region about 800 cm^{-1} corresponding to vibrations of $\text{Ge}-\text{O}^-$ bonds? How can you explain appearance of bands in the region of vibrations of carrying negative charges non-bridging oxygens? Are there negative charges in the GeO_2 melt?

Fig. R8 | (a) A typical $\text{Ge}-\text{O}^-$ stretching vibration of the $[\text{GeO}_2\text{O}_2]^{2-}$ motif in the Li_2GeO_3 crystal and (b) A typical $\text{Ge}-\text{O}$ stretching vibration of the $[\text{GeO}_2]$ motif in the GeO_2 melt.

$[\text{GeO}_2\text{O}_2]^{2-}$ has a characteristic Raman band located around 800 cm^{-1} , which originates from the symmetric stretching vibration of the $\text{Ge}-\text{O}^-$ bond. The case of the Li_2GeO_3 crystal has been reported in our previous paper [Zhang, S. J., Wan, S. M., Zeng, Y., Jiang, S. J., Gong, X. Y. & You, J. L. *Inorg. Chem.* **58**, 5025–5030 (2019)] and is shown in Fig. R8a. Unlike the $[\text{GeO}_2\text{O}_2]^{2-}$ motif in metagermanate crystals/glasses, the $[\text{GeO}_2\text{O}_n]$ motif in the GeO_2 melt is electrically neutral (more strictly speaking, the Ge and O atoms may carry partial charge because of their different electronegativities). The symmetric stretching vibrations of the $\text{Ge}-\text{O}$ bond in $[\text{GeO}_2]$ are located at around 715 cm^{-1} (Fig. R8b). The frequencies are lower than that of the $\text{Ge}-\text{O}^-$ bond. The phenomenon reflects that the $\text{Ge}-\text{O}$ bond is weaker than the $\text{Ge}-\text{O}^-$ bond and the electron density around the $\text{Ge}-\text{O}$ bond is lower than that of the $\text{Ge}-\text{O}^-$ bond.

The relevant content has been added into the main text (Lines 108 to 112).

Reviewer #3 (Remarks to the Author):

This is an interesting computational and experimental paper on the Raman spectra of GeO_2 melts identifying a new type of structure involving a 3-fold coordinated Ge in the melt. The work overall is quite nice, but the presentation needs work. Too much of the critical information is in the SI and a reader should not have to have the SI open to read the paper. In this case, one cannot read the paper without the SI present. In addition, the work would be hard to reproduce given the information in the paper.

1. Line 52. Reword vigorous atomic vibration as this does not make any sense. I would suggest thinking about using the term fluxional bonding as bonds are readily made and broken in the melt. The same is true for line 53 as I have no idea what six vibrant oxygen are. Vibrant is not properly used in this contest as it has a very different meaning.

Fig. R9 | Structural transformations of GeO_2 during heating.

Here, we want to state the fact/view that the coordination number of Ge in GeO_2 decreases as temperature increases. GeO_2 has three different phases at ambient pressure (Fig. R9). Below 1281 K, the stable phase is rutile-type GeO_2 in which the Ge atom is coordinated by six O atoms. From 1281 K to 1388 K (the GeO_2 melting point), the stable phase is quartz-type GeO_2 in which the Ge atom is coordinated by four O atoms. Above 1388 K, the quartz-type phase transforms to the melt phase in which the Ge atom may be coordinated by three O atoms.

Furthermore, we want to point out that the decrease in the coordination number of Ge in GeO_2 is associated with the vibration amplitude (or intensity) of the O atom (Fig. R9). At low temperature, the O atom vibrates with small amplitudes, and thus six O atoms can coordinate with the Ge atom, which corresponds to the GeO_6 motif in rutile-type GeO_2 . At high temperature, the O atoms vibrate with large amplitudes; as a consequence, fewer O atoms are allowed to accommodate in the limited space around the Ge atom. It is the reason why rutile-type GeO_2 will transform to quartz-type GeO_2 (characterized by the GeO_4 motif) at 1281 K and then to the melt (characterized by the GeO_3 motif) at 1388 K.

The relevant content, including Figs. R9, has been added into the main text (Lines 58 to 63 and Fig. 2).

“fluxional bonding” is a great term to describe the melt character. It has been adopted in the revised manuscript.

2. In Figure 2, what is the source of the data for the pictures before and after melting? This needs to be given in the text or figure caption. Figure S2 shows what is going on much better than the current Figure 2 and should be added to it.

The data for the pictures have been added into the caption of Fig. 2 (Lines 51 to 53) and given in the main text (Lines 77 to 80). Fig. 2B has been replaced by Fig. S2 in the revised version.

3. Please show a picture of the Li_2GeO_3 crystal in the text. I think that Figure S1 is critical to the paper and should be in the text.

By a careful comparison, we found that although the Raman bands of the GeO_2 melt are consistent in frequency with that of the Li_2GeO_3 crystal, their corresponding vibrational modes are different. For this reason, Fig. S1 was not adopted in the revised manuscript in order to avoid misunderstanding, namely, the Raman bands of the GeO_2 melt and of the Li_2GeO_3 crystal are wrongly thought to be consistent both in frequency and in vibrational mode.

Fig. S1 has been deleted from the Supplementary Information.

4. Figure S3 is a critical benchmark and should be in the computational methods section in the text.

Fig. S3 has been inserted into the Methods section as Fig. 7.

5. I really do not think that their bond distances in the melt are good to 3 decimal places in angstroms if these are from dynamics type simulations which they need to be at the higher temperature. They are likely to be average values and good to only 2 decimal places.

According to our research, the computational Raman spectra are very sensitive to the structural parameters including the bond lengths. Only when the bond lengths are accurate to three decimal places can the satisfactory spectrum be achieved.

6. The authors note that there are allowed and forbidden IR and Raman bands in Table S2. This can only happen if symmetry is present. Is there symmetry in the chain? This needs to be discussed in more detail. Also, as they have the values, the Raman and IR intensities of all these transitions need to be given.

Yes, the $[\text{GeO}\emptyset_2]_n$ chain model has the orthorhombic D_{2h}^{16} symmetry (space group No.62, $Pnma$). The symmetry has been provided in the revised manuscript (Lines 79 and 80). The .cif file of the chain structure will be available upon request. The intensities of all vibrational modes have been added in Supplementary Table 2.

7. Where are the computed Raman and IR intensities and the associated frequencies for the melt and the pure crystal? These need to be given.

All the computational frequencies and intensities for the $[\text{GeO}\emptyset_2]_n$ chain and for the quartz-type GeO_2 crystal are provided in Supplementary Tables 2 and 3, respectively.

8. The high frequency bands are likely to be Ge–O stretches based on what I know of the bands in metal oxides. Does the number of Ge–O stretches correspond to the numbers of Ge–O bonds in the calculations? How many Ge and O are there in the chain calculations used for Table S2?

Fig. R10 | Four Ge–O stretching modes of the $[\text{GeO}\emptyset_2]_n$ chain model.

The $[\text{GeO}\emptyset_2]_n$ chain model for the spectrum computation is provided in Fig. 3d and Supplementary Fig. 1 and Table 1 for more details. The model consists of two $[\text{GeO}\emptyset_2]_n$ chains; each chain has two $\text{GeO}\emptyset_2$ structural units. According to the DFT computation, there are four Ge–O stretching modes corresponding to the four Ge–O bonds (Figs. R10 and 3d). Among them, the 713 and 718 cm^{-1} modes are Raman active. In Fig. R10, two unit cells are used in order to display the vibrational modes more clearly.

Fig. R10 has been added in the Supplementary Information as Supplementary Fig. 3.

9. Details of the Raman calculations and how the Raman intensities were calculated and corrected need to be in the Supporting Information in much more detail to allow the results to be reproduced. I would think that the line widths seem to be quite narrow for comparison to the experiment in the melt.

The Raman intensity of a vibrational mode can be computed from the derivative of the dielectric polarizability tensor with respect to the mode amplitude. In CASTEP, the dielectric polarizability tensor is computed by the DFPT (density-functional perturbation theory) method. The theoretical and computational details have been described by Porezag and Pederson [Ref. 24: Porezag, D. & Pederson M. R. *Phys. Rev. B*, **54**, 7830–7836 (1996)].

CASTEP can calculate and correct Raman intensities by itself and output a Raman spectrum by specifying a measuring temperature, an incident light wavelength, a line-shape function and a FWHM (full width at half-maximum). All of the parameters have been provided in the Methods section for computing the Raman spectra of the $[\text{GeO}\emptyset_2]_n$ chain and quartz-type GeO_2 .

As we know, it is difficult to assign a reasonable FWHM for a specified vibrational mode at a given temperature. Therefore, CASTEP often uses a uniform FWHM to broaden all Raman lines, and thus it is hard to obtain a computational spectrum in perfect agreement with the experimental one. In this work, all FWHMs are set to 10 cm^{-1} . Indeed, the line width is much narrower than the experimental value. However, with the FWHM, not only all of the weak bands are observable, but also the intensities of all bands are displayed clearly. A similar manner has also been used for simulating the Raman spectra of metastable germanium [Zhao, Z. S. et al. *Nat. Commun.* **8**, 13909 (2017)], sodium polyhydrides [Struzhkin, V. V. et al.

Nat. Commun. **7**, 12267 (2016)], etc.

10. I am not sure how relevant GeO₂ chemistry is to SiO₂ chemistry as stated in the paper as the Si–O bonds are stronger than the Ge–O bonds so will require higher temperatures. Do they expect to find the same structures in SiO₂ melts? As this is discussed in the conclusions, the authors need to comment on the relevance of this to the actual SiO₂ chemistry they want to mimic for modeling the Earth’s core.

Phase transformations involving changes in Si coordination are of great interest in geochemistry. However, experimental studies of these phase transformations are difficult because they usually occur under extreme temperature and pressure conditions. Therefore, germanates are often used as models for silicates because they exhibit similar transformations to silicates at lower pressures [Ringwood, A. E. & Seabrook, M. J. *Geophys. Res.* **68**, 4601–4609 (1963); Ross, N. L, Akaogi, M., Navrotsky, A., Suzuki, J. I. & McMillan, P. J. *Geophys. Res.* **91**, 4685–4696 (1986)].

Vitreous SiO₂ is usually regarded as the supercooled state of molten SiO₂, and thus has the SiO₂ melt structural features. The Raman spectrum of vitreous SiO₂, reported by Dovbeshko et al., is shown in Fig. R11b [Dovbeshko, D., et al. *IOP Conf. Ser.: Mater. Sci. Eng.* **38**, 012008 (2012)]. In comparison to the GeO₂ melt spectrum (Fig. 11a), all the vitreous SiO₂ Raman bands, except that located around 600 cm⁻¹, have their corresponding bands in the GeO₂ melt spectrum. We thus infer that vitreous SiO₂, as well as molten SiO₂, has a similar structure to the GeO₂ melt. That is, the SiO₂ glass/melt probably contains the [SiO₂]_n motif. However, this inference needs verifying. The relevant work is ongoing.

The relevant content has been added into the main text (Lines 140 to 143).

Fig. R11 | The Raman spectra of (a) the GeO₂ melt, (b) the SiO₂ glass, and (c) α-quartz.

Overall, some nice work but the paper is too terse, and the work cannot readily be reproduced without more computational details. Figures from the SI need to be moved to the text.

More computational details are provided in the revised version. The relevant figures have been moved to the text from the Supplementary Information.

REVIEWERS' COMMENTS (second round)

Reviewer #1 (Remarks to the Author):

I have read carefully the revised version of the article: "Threefold coordinated germanium in a GeO₂ melt" by Songming Wan et al. The resubmitted version of this article represents a significant improvement with respect to the original version. The authors spent a noticeable effort in addressing all the reviewers' requests. Most of the issues raised by the referees got a proper answer. To my opinion, the discussion, the quality of artworks, the relevance of results presented by the authors fulfill the standard of the journal. I therefore recommend publication of the article, subject to an explanation/change related to the discussion of the electronic structure of GeO₂ planar arrays.

Let's focus on the bottom left panel of Figure 6, and the related discussion in the text. The authors claim that there is no π bond between germanium and oxygen. That is true and is not a novelty. However, the analysis of the bottom left panel of Figure 6 suggests to me that a π structure exists. Indeed, it seems to me that π orbitals are present on all oxygen atoms. More, the material provided by the authors suggest that they are quite close to the Fermi level, another typical property of non-bonding states. I encourage the authors to spend more attention on this hypothesis: there is no π Ge–O bonding, but there is a non-bonding π structure involving oxygen atoms. If this is the case, further (but minor) changes to the text are needed.

Reviewer #2 (Remarks to the Author):

Dear Professor Wan,

Thank you for your valuable comments.

1. As far as we know, the breaking of B–O–B, Ge–O–Ge and Si–O–Si bonds in melts is a natural process. Obviously, the rate of non-bridging oxygen formation should increase with temperature. However, when one Ge–O–Ge bond breaks, one non-bridging oxygen atom carrying a -1 charge and one germanium atom with a charge of $+1$ are formed. Such a state is unbeneficial for the system, resulting in the formation of a new bond. The breaking of bonds of this type occurs constantly and rapidly along with the formation of new ones. The structure referred to by the authors is impossible from the point of view of electroneutrality. A non-bridging oxygen atom carries a negative charge, which is not compensated by anything. At the same time, somewhere nearby there is a positively charged unit $[\text{GeO}_3]^+$ that must interact with something. There are no obstacles to such interactions. However, if we are talking about breaking bonds with further evaporation, then indeed we would expect to observe the eventual formation of GeO₂ or GeO oxides. But this involves another story about double bonds...
2. In connection with remark 1, please describe the reaction mechanism that causes the breaking of the Ge–O–Ge bond. Could you explain how a non-bridging oxygen atom is formed from your point of view?
3. Can you depict all the electrons and atomic orbitals that are involved in the formation of the structural fragment $[\text{GeO}_2]$? What does the hybridization of an oxygen atom comprise in such a fragment?
4. Is the main unit of the proposed structure triangular? Is its structure similar to that of planar triangles $[\text{BO}_2]^-$? Or maybe to the structure of a $[\text{CO}_3]^{2-}$ ion? Please explicate all the chemical bonds due to

which the formation of the $[\text{Ge}\emptyset_2\text{O}]$ molecule occurs. It is unclear how this corresponds to the presence of 6 valence electrons of the oxygen atom.

5. The claim that the chain is electroneutral appears to be contradictory. The bands in the region of $800\text{--}1100\text{ cm}^{-1}$ on the Raman spectrum are characteristic of vibrations of the bonds of the glass-forming cation with non-bridging oxygen atoms carrying negative charges. These are characteristic bands for Q^3 , Q^2 , Q^1 and Q^0 -species (Furukawa and White, 1991; Fuss et al., 2006; Mysen et al., 1982a; Mysen et al., 1980; Mysen et al., 1982b). On this basis, to compare the vibrations of the structural units Q^2 , whose structure can be described as $[\text{Ge}\emptyset_2\text{O}_2]^{2-}$, with the vibrations of the motif $[\text{Ge}\emptyset_2\text{O}]$, seems incorrect. What they have in common are two bridging oxygen atoms. Their difference is precisely in the charge of non-bridging oxygen atoms. As you will be aware, non-bridging oxygen atoms affect the interatomic distance and, as a consequence, the oscillation frequency.
6. The vibrations of symmetrical bonds are characterized by bands in the range from 300 to 650 cm^{-1} . We observe them both in glass and in the melt. Changing the shape of the bands of the Raman spectrum in this area can help to define the structure. Would you like to try a curve-fitting of the low-frequency region of the spectrum on the superposition of Gaussians? The theory of the existence of rings of various sizes C2-C6 can be applied (Henderson and Fleet, 1991a). Perhaps this will explain the changes in the melt network without breaking the bonds? It would be preferable to take both theories into account.
7. TO/LO bands do shift with temperature and with changes in composition (Henderson and Amos, 2003; Henderson and Fleet, 1991b; Verweij and Buster, 1979). This seems impossible when comparing the spectra of melts with the crystal spectrum. Perhaps it would be productive to compare the spectra of glass and melts.

Following correction, the manuscript can make a very useful contribution to structural chemistry. It would be interesting to simulate further heating and evaporation. It is possible to compare the obtained data with the results of high-temperature mass spectrometry. Germanate and borate systems are very suitable models for this, since their evaporation occurs at lower temperatures than silicate ones.

References

1. Furukawa, T. and W. B. White. 1991. Raman spectroscopic investigation of the structure of silicate glasses. IV. Alkali-silico-germanate glasses. *J. Chem. Phys.* 95: 776–784.
2. Fuss, T. A., Mogus-Milankovic, C. S., Ray, C. E. Leshner, R. Youngman, and D. E. Day. 2006. Ex situ XRD, TEM, IR, Raman and NMR spectroscopy of crystallization of lithium disilicate glass at high pressure. *J. Non-Cryst. Solids.* 352: 4101–4111.
3. Henderson, G. S. and R. T. Amos 2003. The structure of alkali germanophosphate glasses by Raman spectroscopy. *J. Non-Cryst. Solids.* 328: 1–19.
4. Henderson, G. S. and M. E. Fleet. 1991a. The structure of alkali germanate and silicate glasses by Raman spectroscopy. *Trans. Am. Crystallogr. Assoc.* 27: 269–278.
5. Henderson, G. S. and M. E. Fleet. 1991b. The structure of glasses along the $\text{Na}_2\text{O}\text{--}\text{GeO}_2$ join. *J. Non-Cryst. Solids.* 134: 259–269.
6. Mysen, B. O., L. W. Finger, D. Virgo, and F. A. Seifert. 1982a. Curve-fitting of Raman spectra of silicate glasses. *Am. Mineral.* 67: 686–695.

7. Mysen, B. O., D. Virgo, and C. M. Scarfe. 1980. Relations between the anionic structure and viscosity of silicate melts—a Raman spectroscopic study. *Am. Mineral.* 65: 690–710.
8. Mysen, B. O., D. Virgo and F. A. Seifert. 1982b. The structure of silicate melts: Implications for chemical and physical properties of natural magma. *Rev. Geophys.* 20: 353–383.
9. Verweij, H., and J. H. J. M. Buster. 1979. The structure of lithium, sodium and potassium germanate glasses, studied by Raman scattering. *J. Non-Cryst. Solids.* 34: 81–99.

Reviewer #3 (Remarks to the Author):

There are still issues with the manuscript.

Response to reviewer 1.

The density of states does not provide any description of the bonding.

Response to reviewer 3.

Comment 5. The authors have a serious error if the Raman calculation require 3 decimal place accuracy in angstroms for good calculations. This is not correct, and this is well established for decades of calculations in the literature. There is something very wrong with what they are doing if this is what is required in terms of the geometry.

Lines 88 to 91 are new and make absolutely no chemical sense. A change in 0.03 eV per unit is only 0.7 kcal/mol and this is very small relive to the error in the calculations of ± 3 to 5 kcal/mol at best. This does prove anything in terms of accuracy. Lines 117 to 124. The statements that the bonding orbitals on Ge are the 4s and 4p is trivial and obvious. How do they know if they are bonding from the DOS?

I think that the authors need to restate the electronic structure description in terms of the percent ionic character in the bonds. One does not expect a strong Ge–O pi bond here just as they find.

If they want to publish the SiO₂ results, they should do so. Remove the preliminary studies comments in the conclusions there are no details (Lines 142–143). If they want to make the comparison between SiO₂ and GeO₂ for geochemical applications, the response given for comment 10 is nice and should be in the text. What is there is not useful, as it is too terse. Figure R11 should be somewhere in the manuscript if they want to make such comments about SiO₂.

Pleas provide in the text a comment/benchmark on the quality of the Wu-Cohen GGA as it not routinely used by many chemists.

For the chain optimization, what unit cell was used then? Line 165.

Explain in the text how the Raman spectra were calculated and give the FWHM value in the text.

Table S2 is clearly incorrect. There cannot be bands with 0 cm⁻¹ values. They have not projected something out correctly. The same is true for Table S3.

Please give IR intensities in the normal units of km/mol to one decimal place at most. Is one of the structures centrosymmetric so the Raman/IR exclusion rule applies?

Response to the comments from Reviewers (second round)

We sincerely thank all the reviewers for the constructive comments on our manuscript entitled "Threefold coordinated germanium in a GeO₂ melt" (NCOMMS-22-43430). The Main Text and the Supplementary Information have been revised according to the comments. The point-to-point responses are as follows:

Reviewer #1 (Remarks to the Author):

I have read carefully the revised version of the article: "Threefold coordinated germanium in a GeO₂ melt" by Songming Wan et al. The resubmitted version of this article represents a significant improvement with respect to the original version. The authors spent a noticeable effort in addressing all the reviewers' requests. Most of the issues raised by the referees got a proper answer. To my opinion, the discussion, the quality of artworks, the relevance of results presented by the authors fulfill the standard of the journal. I therefore recommend publication of the article, subject to an explanation/change related to the discussion of the electronic structure of GeO₂ planar arrays.

Let's focus on the bottom left panel of Figure 6, and the related discussion in the text. The authors claim that there is no π bond between germanium and oxygen. That is true and is not a novelty. However, the analysis of the bottom left panel of Figure 6 suggests to me that a π structure exists. Indeed, it seems to me that π orbitals are present on all oxygen atoms. More, the material provided by the authors suggest that they are quite close to the Fermi level, another typical property of non-bonding states. I encourage the authors to spend more attention on this hypothesis: there is no π Ge-O bonding, but there is a non-bonding π structure involving oxygen atoms. If this is the case, further (but minor) changes to the text are needed.

We appreciate the reviewer's positive evaluation on our work.

The total and projected densities of states of [GeO \emptyset_2]_n (Fig. 5a) indicate that the orbital in the energy range from -0.2 to 0.0 eV mainly derives from the O-2p atomic orbitals; namely, the orbital shown in the bottom left panel of Fig. 6 is a non-bonding O-2p atomic orbital rather than a non-bonding π structure.

The orbital nature has been described in the Main Text (Lines 124, 131 and 133).

Reviewer #2 (Remarks to the Author):

1. As far as we know, the breaking of B-O-B, Ge-O-Ge and Si-O-Si bonds in melts is a natural process. Obviously, the rate of non-bridging oxygen formation should increase with temperature. However, when one Ge-O-Ge bond breaks, one non-bridging oxygen atom carrying a -1 charge and one germanium atom with a charge of +1 are formed. Such a state is unbeneficial for the system, resulting in the formation of a new bond. The breaking of bonds of this type occurs constantly and rapidly along with the formation of new ones. The structure referred to by the authors is impossible from the point of view of electroneutrality. A non-bridging oxygen atom carries a negative charge, which is not compensated by anything. At the same time, somewhere nearby there is a positively charged unit [GeO₃]⁺ that must interact with something. There are no obstacles to such interactions. However, if we are talking about breaking bonds with further evaporation, then indeed we would expect to observe the eventual formation of GeO₂ or GeO oxides. But this involves another story about double bonds...

Fig. R1 | Mulliken charge distribution in the $[\text{GeO}\emptyset_2]_n$ chain.

As Ge–O–Ge bonds break, the electrons will be redistributed between Ge and O in terms of their electronegativities and eventually reach a steady state. The DFT computational result shows that the Mulliken charges of Ge, non-bridging O and bridging O in the $[\text{GeO}\emptyset_2]_n$ chain are stabilized at +1.83 e, –0.90 e and –0.93 e, respectively (Fig. R1), which indicates that all the Ge–O bonds in the chain are polar, and the whole chain is electroneutral (the positive charge on the Ge atom is compensated by the negative charge on the neighboring O atoms).

Fig. R1 has been added in the Supplementary Information as Supplementary Fig. 4a. The relevant discussion has been added into the Main Text (Lines 150 to 153).

2. In connection with remark 1, please describe the reaction mechanism that causes the breaking of the Ge–O–Ge bond. Could you explain how a non-bridging oxygen atom is formed from your point of view?

The breaking of the Ge–O–Ge bond is associated with the vibration of the O atom. At high temperature, the O atom vibrates with larger amplitude and thus fewer O atoms can be accommodated in the limited space around the Ge atom. As a result, the $[\text{Ge}\emptyset_4]$ motif (four O atoms around the Ge atom) transforms to the $[\text{GeO}\emptyset_2]$ motif (three O atoms around the Ge atom); namely, a Ge–Ø bond in the $[\text{Ge}\emptyset_4]$ motif breaks and yields a non-bridging O atom and a threefold coordinated Ge atom.

The relevant content is provided in the Main Text (Fig. 2 and Lines 55 to 65).

3. Can you depict all the electrons and atomic orbitals that are involved in the formation of the structural fragment $[\text{GeO}\emptyset_2]$? What does the hybridization of an oxygen atom comprise in such a fragment?

Ge- $4s^2 4p^2$ and O- $2s^2 2p^4$ orbitals and electrons are involved in the process. According to the densities of states of the $[\text{GeO}\emptyset_2]_n$ chain (Fig. 5a), the orbital hybridization of O-2s with O-2p is insignificant in the formation.

The valence electronic configurations have been added in the Main Text (Lines 119 and 120).

4. Is the main unit of the proposed structure triangular? Is its structure similar to that of planar triangles $[\text{B}\emptyset_2\text{O}]^-$? Or maybe to the structure of a $[\text{CO}_3]^{2-}$ ion? Please explicate all the chemical bonds due to which the formation of the $[\text{Ge}\emptyset_2\text{O}]$ molecule occurs. It is unclear how this corresponds to the presence of 6 valence electrons of the oxygen atom.

Yes, the proposed $[\text{GeO}\emptyset_2]$ motif is triangular and formed by one Ge atom, one non-bridging O atom and two bridging O atoms. Its geometry is more similar to that of $[\text{BO}\emptyset_2]^-$. However, unlike $[\text{BO}\emptyset_2]^-$, which is negatively charged, the $[\text{GeO}\emptyset_2]$ motif as a whole is electroneutral.

Fig. R2 | Hybridization and bonding in the $[\text{GeO}_{0.2}]$ motif. a Hybridization scheme for the germanium atom. **b** Interactions between $\text{Ge-}sp^2$ and $\text{O-}2p$ orbitals.

The electronic densities of states of the $[\text{GeO}_{0.2}]_n$ chain indicate that the bonding orbitals of Ge have some hybrid character. The trigonal planar geometry of the $[\text{GeO}_{0.2}]$ motif suggests that Ge in $[\text{GeO}_{0.2}]$ is the sp^2 hybridization. In the hybridization, the $\text{Ge-}4s$ orbital mixes with two $\text{Ge-}4p$ orbitals to form three $\text{Ge-}sp^2$ (for example, $4s4p_x4p_y$) orbitals of equal energy. At the same time, a $\text{Ge-}4s$ electron is excited to the empty $\text{Ge-}4p$ orbital and results in the $4s^14p_x^14p_y^14p_z^1$ configuration (Fig. R2a). The three $\text{Ge-}sp^2$ orbitals overlap with three half-filled $2p$ orbitals on three different oxygen atoms, forming three σ bonds ($\text{Ge-O}/\text{Ge-O}$ bonds). The remaining perpendicular $4p_z^1$ orbital is not involved in the bonding (Fig. R2b), but can interact weakly with two neighboring half-filled $2p$ -orbital on two non-bridging oxygen atoms in two adjacent $[\text{GeO}_{0.2}]_n$ chains, as shown in the second image of Fig. 7a.

The oxygen atom, with valence electron configuration $2s^22p^4$, has two half-filled $2p$ -orbitals. For the bridging oxygen atom, the two half-filled orbitals overlap with two neighboring $\text{Ge-}sp^2$ orbitals to form two Ge-O bonds (σ bonds). For the non-bridging oxygen atom, only one half-filled orbital overlaps with a neighboring $\text{Ge-}sp^2$ orbital to form the Ge-O bond (σ bond); the remaining half-filled $2p$ -orbital is not involved in the bonding, but can interact weakly with two neighboring $\text{Ge-}4p_z^1$ orbitals on two different $[\text{GeO}_{0.2}]_n$ chains, as shown in the second image of Fig. 7a.

Fig.R2 and the relevant discussion have been added in the Main Text (Fig. 6, Lines 127 to 133, and Lines 140 to 144).

5. The claim that the chain is electroneutral appears to be contradictory. The bands in the region of $800\text{--}1100\text{ cm}^{-1}$ on the Raman spectrum are characteristic of vibrations of the bonds of the glass-forming cation with non-bridging oxygen atoms carrying negative charges. These are characteristic bands for Q^3 , Q^2 , Q^1 and Q^0 – species (Furukawa and White, 1991; Fuss et al., 2006; Mysen et al., 1982a; Mysen et al., 1980; Mysen et al., 1982b). On this basis, to compare the vibrations of the structural units Q^2 , whose structure can be described as $[\text{GeO}_{2.0}]^{2-}$, with the vibrations of the motif $[\text{GeO}_{2.0}]$, seems incorrect. What they

have in common are two bridging oxygen atoms. Their difference is precisely in the charge of non-bridging oxygen atoms. As you will be aware, non-bridging oxygen atoms affect the interatomic distance and, as a consequence, the oscillation frequency.

Fig. R3 | Mulliken charge distribution in the $[\text{GeO}\emptyset_2]_n$ chain and the $[\text{GeO}_2\emptyset_2^{2-}]_n$ chain. a The $[\text{GeO}\emptyset_2]_n$ chain. **b** The $[\text{GeO}_2\emptyset_2^{2-}]_n$ chain.

In this paper, we compare the 715 cm^{-1} band of the $[\text{GeO}\emptyset_2]_n$ chain with the 815 cm^{-1} band of the $[\text{Ge}\emptyset_2\text{O}_2^{2-}]_n$ (Q^2) chain because both of them arise from the stretching vibration of the Ge–O bond and because the Ge–O bonds in the two chains are the same type of chemical bond. Mulliken charge population analysis reveals that, in the two chains, all the Ge atoms are positively charged, all the O atoms are negatively charged, and all the Ge–O bonds are polar (see Fig. R3. The Mulliken charges of Ge, non-bridging O and bridging O are +1.13 e, –1.18 e and –1.03 e, respectively, in the $[\text{GeO}_2\emptyset_2^{2-}]_n$ chain, and +1.83 e, –0.90 e and –0.93 e, respectively, in the $[\text{GeO}\emptyset_2]_n$ chain).

According to Hooke's law, the Ge–O bond strength is positively correlated to its stretching frequency. The Ge–O stretching vibration of $[\text{GeO}\emptyset_2]_n$ has a lower frequency, reflecting that the Ge–O bond in $[\text{GeO}\emptyset_2]_n$ is weaker. Thanks for the referee's reminding, we checked the Ge–O bond lengths in the two chains. The bond length data (1.84 Å in $[\text{GeO}\emptyset_2]_n$ and 1.733 Å in $[\text{Ge}\emptyset_2\text{O}_2^{2-}]_n$, see Fig. R3) also support our inference.

The above data and discussion are provided in the Main Text (Lines 150 to 156) and the Supplementary Information (Supplementary Fig. 4 and the relevant discussion).

6. The vibrations of symmetrical bonds are characterized by bands in the range from 300 to 650 cm^{-1} . We observe them both in glass and in the melt. Changing the shape of the bands of the Raman spectrum in this area can help to define the structure. Would you like to try a curve-fitting of the low-frequency region of the spectrum on the superposition of Gaussians? The theory of the existence of rings of various sizes C2-C6 can be applied (Henderson and Fleet, 1991a). Perhaps this will explain the changes in the melt network without breaking the bonds? It would be preferable to take both theories into account.

Fig. R4 | A simulated Raman spectrum of the GeO₂ melt (black curve). The melt consists of the [GeO \emptyset_2]_n chain and the [Ge \emptyset_4]_n network. Red curve is the contribution from the [GeO \emptyset_2]_n chain; blue curve is the contribution from the [Ge \emptyset_4]_n network.

The black curve in Fig. R4 is a Raman spectrum of the GeO₂ melt simulated by the Lorentzian line-shape function (the Lorentzian line-shape function is better than the Gaussian line-shape function for Raman curve fitting in this work). The spectrum resembles the experimental spectrum (at 1,500 K). But we do not think the result is trusted because too many parameters, including the line-shape function, the ratio of [GeO \emptyset_2] to [Ge \emptyset_4] and the FWHM of each Raman band, can be manipulated to get a spectrum that looks good. Therefore, curve-fitting was not applied in this work.

The viewpoint that GeO₂ melt/glass contains small-membered rings was proposed by Henderson et al., which is based on the band assignments for the SiO₂ melt/glass (an analogue of the GeO₂ melt). The viewpoint has been widely cited for decades. However, Henderson et al. also recognized that the band assignments are tentative and require further studied [Henderson, G. S., Bancroft, G. M., Fleet, M. E. & Rogers, D. J. *Am. Mineral.* **70**, 946–960 (1985)]. They once said, “The question of whether or not the medium-range structure of GeO₂ glass consists of six- or four-membered rings remains unanswered and open for further studies.” [Ref. 1: Micoulaut, M., Cormier, L. & Henderson, G. S. *J. Phys.: Condens. Matter.* **18**, R753–R784 (2006)]

The Si–O–Si bond breaks in the SiO₂ melt [Sharma, S. K., Mammone, J. F. & Nicol, M. F. *Nature* **292**, 140–141 (1981); Stebbins, J. F. & Farnan, I. *Science* **255**, 586–589 (1992)]. As an analogue, the Ge–O–Ge bond should break in the GeO₂ melt. On the basis of the inference, we constructed the [GeO \emptyset_2]_n chain model. Using the structural model, all the Raman bands of the GeO₂ melt are interpreted. Thus, the ring structures are not taken into account in this paper.

7. TO/LO bands do shift with temperature and with changes in composition (Henderson and Amos, 2003; Henderson and Fleet, 1991b; Verweij and Buster, 1979). This seems impossible when comparing the spectra of melts with the crystal spectrum. Perhaps it would be productive to compare the spectra of glass and melts.

Fig. R5 | Raman spectra of the GeO₂ crystal and glass at room temperature and Raman spectra of the GeO₂ crystal and melt near the melting point.

Sharma et al. have reported the Raman spectra of the GeO₂ crystal and glass at room temperature and the Raman spectra of the GeO₂ crystal and melt near the melting point [Sharma, S. K., Cooney, T. F., Wang, Z. F. & van der Laan, S. *J. Raman Spectrosc.* **28**, 697–709 (1997)]. The results are shown in Fig. R5. The TO/LO bands of the crystal and glass are both located around 860 and 970 cm⁻¹ at room temperature. After melting, the two bands shift to around 820 and 940 cm⁻¹. The results are in agreement with our experimental results. In the Raman spectrum of the GeO₂ melt recorded at 1,500 K (Fig. 4 in the Main Text), two Raman bands located at around 820 and 940 cm⁻¹ are observable although they are very weak. The result clearly indicates that the Raman shifts of the TO/LO bands are limited as temperature increases. The 750 and 810 cm⁻¹ bands of the GeO₂ melt do not correspond to the 860 and 970 cm⁻¹ bands of the crystal.

Following correction, the manuscript can make a very useful contribution to structural chemistry. It would be interesting to simulate further heating and evaporation. It is possible to compare the obtained data with the results of high-temperature mass spectrometry. Germanate and borate systems are very suitable models for this, since their evaporation occurs at lower temperatures than silicate ones.

Thanks again for the reviewer's constructive comments.

Reviewer #3 (Remarks to the Author):

1. The density of states does not provide any description of the bonding.

The bonding of the [GeO₂] motif has been described in the response to the fourth comment of Reviewer 2. The description has been added into the Main Text (Fig. 6 and Lines 127 to 133).

2. Comment 5. The authors have a serious error if the Raman calculation require 3 decimal place accuracy in angstroms for good calculations. This is not correct, and this is well established for decades of

calculations in the literature. There is something very wrong with what they are doing if this is what is required in terms of the geometry

Fig. R6 | Computational Raman spectra of two $[\text{GeO}\emptyset_2]_n$ chain models with different Ge-O/Ge- \emptyset bond lengths. a $d_{\text{Ge-O}} = 1.844 \text{ \AA}$, $d_{\text{Ge-}\emptyset_1} = 1.783 \text{ \AA}$, and $d_{\text{Ge-}\emptyset_2} = 1.770 \text{ \AA}$. **b** $d_{\text{Ge-O}} = 1.837 \text{ \AA}$, $d_{\text{Ge-}\emptyset_1} = 1.780 \text{ \AA}$, and $d_{\text{Ge-}\emptyset_2} = 1.769 \text{ \AA}$.

An alternative unit cell, with parameters $a = 4.90 \text{ \AA}$, $b = 3.87 \text{ \AA}$ and $c = 8.10 \text{ \AA}$, was employed to compute the Raman spectrum of the $[\text{GeO}\emptyset_2]_n$ chain. The computational spectrum (red curve in Fig. R6) is very similar to that in Fig. 4a (black curve in Fig. R6). The three Ge-O/Ge- \emptyset bonds in the $[\text{GeO}\emptyset_2]$ motif are 1.837 \AA , 1.780 \AA and 1.769 \AA in length, different from the values present in the Main Text in the third decimal place (1.844 \AA , 1.783 \AA and 1.770 \AA in the unit cell with parameters $a = 4.85 \text{ \AA}$, $b = 3.82 \text{ \AA}$ and $c = 8.15 \text{ \AA}$). The result supports the referee's comment.

Following the comment, the accuracy of the unit cell parameters and bond lengths for the $[\text{GeO}\emptyset_2]_n$ chain model has been modified to the second decimal place in the revised manuscript (Lines 79, 80, 85, 86 and 87) and the Supplementary Information (Supplementary Fig. 1).

3. Lines 88 to 91 are new and make absolutely no chemical sense. A change in 0.03 eV per unit is only 0.7 kcal/mol and this is very small relative to the error in the calculations of ± 3 to 5 kcal/mol at best. This does prove anything in terms of accuracy. Lines 117 to 124. The statements that the bonding orbitals on Ge are the 4s and 4p is trivial and obvious. How do they know if they are bonding from the DOS?

Chemical bonding in the $[\text{GeO}\emptyset_2]_n$ chain results from the overlap of two separate orbitals on the Ge and O atoms, and thus every bonding orbital must be the mixture of the Ge and O atomic orbitals. The contribution of the Ge and O atomic orbitals to the bonding orbitals can be described by the DOS of $[\text{GeO}\emptyset_2]_n$. In the TDOS (total DOS) curve, every bonding orbital has a definite energy range. If the TDOS in the energy range is composed of the PDOSs (projected DOSs) of Ge and O, the Ge atom is considered to be bonded with the O atom.

4. I think that the authors need to restate the electronic structure description in terms of the percent ionic character in the bonds. One does not expect a strong Ge-O pi bond here just as they find.

In CASTEP, the covalent/ionic character of a chemical bond is assessed by atomic charge and bond population. For the $[\text{GeO}\emptyset_2]_n$ chain, the Mulliken charges of Ge, non-bridging O and bridging O are $+1.83 e$, $-0.90 e$ and $-0.93 e$, respectively. The bond populations of Ge-O ($d_{\text{Ge-O}} = 1.84 \text{ \AA}$), Ge- \emptyset_1 ($d_{\text{Ge-}\emptyset_1} = 1.78 \text{ \AA}$)

and Ge- \emptyset 2 ($d_{\text{Ge}-\emptyset 2} = 1.77 \text{ \AA}$) are +0.53 e, +0.55 e and +0.53 e, respectively (Fig. R1). The above result reveals the polar covalent nature of the Ge-O/Ge- \emptyset bonds.

The data and discussion are provided in the Main Text (Lines 150 to 154) and in the Supplementary Information (Supplementary Fig. 4 and the relevant discussion).

5. If they want to publish the SiO₂ results, they should do so. Remove the preliminary studies comments in the conclusions there are no details (Lines 142–143). If they want to make the comparison between SiO₂ and GeO₂ for geochemical applications, the response given for comment 10 is nice and should be in the text. What is there is not useful, as it is too terse. Figure R11 should be somewhere in the manuscript if they want to make such comments about SiO₂.

Since it is only a preliminary study on the SiO₂ melt structure and inappropriate for publishing in the present state, we decide to delete the comments on the SiO₂ melt structure.

6. Please provide in the text a comment/benchmark on the quality of the Wu-Cohen GGA as it not routinely used by many chemists.

GGA-WC functional was proposed and used to describe the exchange and correlation effects by Wu and Cohen in 2006 [Ref. 24: Wu, Z. & Cohen, R. E. *Phys. Rev. B* **73**, 235116 (2006)]. It is one latest GGA functional and significantly improves the computational accuracy relative to the most popular GGA-PBE functional.

The comment has been added in the Methods section (Lines 176 to 178).

7. For the chain optimization, what unit cell was used then? Line 165.

The unit cell parameters have been added in the Methods section (Lines 185 and 186).

8. Explain in the text how the Raman spectra were calculated and give the FWHM value in the text.

CASTEP uses density functional perturbation theory (DFPT, also known as the linear response method) to compute the Raman spectra (modes, frequencies and intensities at the center of the Brillouin zone (Γ point)) of quartz-type GeO₂ and the [GeO \emptyset ₂]_n chain. By constructing the Hessian matrix, Raman vibrational information was obtained. The eigenvectors of the matrix are the Raman modes; the square roots of the eigenvalues are the Raman frequencies. The intensity of each vibrational mode was computed from the derivative of the dielectric polarizability tensor with respect to the mode amplitude. Computational Raman intensities were corrected by the Bose-Einstein factors calculated with the excitation source wavelength (355 nm) and the experimental temperatures (300 K for quartz-type GeO₂ and 1,500 K for the [GeO \emptyset ₂]_n chain). The computational spectra have been broadened using the Lorentzian line-shape function with a fixed full width at half-maximum (FWHM) of 10 cm⁻¹.

The computational method has been added in the Methods section (Lines 187 to 192). The FWHM value is provided in Line 196.

9. Table S2 is clearly incorrect. There cannot be bands with 0 cm⁻¹ values. They have not projected something out correctly. The same is true for Table S3.

For a three-dimensional periodic model, there are 3N vibrational modes at the Γ point. The three lowest-frequency modes, called acoustic mode, are attributed to the pure translation of unit cells (all atoms move essentially in phase in an acoustic mode), and thus they are modes of zero frequency.

10. Please give IR intensities in the normal units of km/mol to one decimal place at most. Is one of the structures centrosymmetric so the Raman/IR exclusion rule applies?

Following the suggestion, infrared intensities are given in the normal units (km/mol) and to one decimal place.

The $[\text{GeO}\text{O}_2]_n$ chain model is a centrosymmetric structure (space group $Pnma$), and thus no band can be active in both Raman scattering and infrared absorption (Raman/IR exclusion rule).

REVIEWERS' COMMENTS (third round)

Reviewer #1 (Remarks to the Author):

I read the revised version of the article "Threefold coordinated germanium in a GeO₂ melt" by Songming Wan et al. I consider positively the efforts of the author in improving the manuscript with respect to the original version. To my opinion, this article contains several useful hints for the comprehension of "odd" atomic arrangements within the GeO₂ melt, which is quite intriguing and useful for geologists.

The article is quite clear and concise. The quality of English is appropriate. The quality of artworks is appropriate as well. With this respect, the article might deserve publication in this journal.

However, considering entirely the referee's comments, I do believe that the authors must spend some more efforts before this paper be published. Indeed, one of the key elements of this study is the electronic structure of three-fold coordinated Ge. What about the redistribution of electrons within the chain? What about their localization, and bonding? Are Pi electrons atomically localized on oxygen? The scientific community set up several theoretical tools to answer properly these questions. The Quantum Theory of Atoms in Molecule, the Electron Localization Function, among others, are powerful instruments of analysis, and provide answers not basis set dependent. Conversely, we do know that the Mulliken scheme is useful just to provide a first sight picture of the bonding structure, but present several serious shortcomings (is basis set dependent, and Mulliken populations have no formal and unique physical basis). I therefore encourage the authors to carry out the analysis of the electronic structure of the system under investigation using an instrument of analysis more grounded in physics. I'm confident that this would add the article a much more relevant physical meaning, such as to catch the interest of a wide community of scientists.

This comment applies for all sections where the electronic structure is discussed within the text.

Reviewer #3 (Remarks to the Author):

The authors have addressed the issues I raised in a satisfactory manner. Accept as is.

Response to the comments from Reviewers (third round)

We sincerely thank you for the valuable comments on our manuscript entitled "Threefold coordinated germanium in a GeO₂ melt" (NCOMMS-22-43430). The Main Text and the Supplementary Information have been revised according to the comments. The point-to-point responses are as follows:

Reviewer #1 (Remarks to the Author):

I read the revised version of the article "Threefold coordinated germanium in a GeO₂ melt" by Songming Wan et al. I consider positively the efforts of the author in improving the manuscript with respect to the original version. To my opinion, this article contains several useful hints for the comprehension of "odd" atomic arrangements within the GeO₂ melt, which is quite intriguing and useful for geologists.

The article is quite clear and concise. The quality of English is appropriate. The quality of artworks is appropriate as well. With this respect, the article might deserve publication in this journal.

However, considering entirely the referee's comments, I do believe that the authors must spend some more efforts before this paper be published. Indeed, one of the key elements of this study is the electronic structure of three-fold coordinated Ge. What about the redistribution of electrons within the chain? What about their localization, and bonding? Are Pi electrons atomically localized on oxygen? The scientific community set up several theoretical tools to answer properly these questions. The Quantum Theory of Atoms in Molecule, the Electron Localization Function, among others, are powerful instruments of analysis, and provide answers not basis set dependent. Conversely, we do know that the Mulliken scheme is useful just to provide a first sight picture of the bonding structure, but present several serious shortcomings (is basis set dependent, and Mulliken populations have no formal and unique physical basis). I therefore encourage the authors to carry out the analysis of the electronic structure of the system under investigation using an instrument of analysis more grounded in physics. I'm confident that this would add the article a much more relevant physical meaning, such as to catch the interest of a wide community of scientists.

Fig. R1 | valence ELF maps for the two structures present in the GeO₂ melt. a The [GeO_{0.5}]_n chain. **b** The [Ge_{0.5}]_n network. The maps are along the planes on which the Ge–O/Ge–Ø bonds lie (the same below).

The electron localization function (ELF) is a measure of the probability of finding an electron pair in a space region, and is an intuitive tool to identify the character of a chemical bond [Becke, A. D. & Edgecombe, K. E. A simple measure of electron localization in atomic and molecular systems. *J. Chem. Phys.* **92**, 5397–5403 (1990)]. In the revised version, the valence ELF was used to analyze the Ge–O/Ge–Ø bonds in the [GeO_{0.5}]_n chain and the [Ge_{0.5}]_n network. Fig. R1a shows a two-dimensional ELF map of

$[\text{GeO}\emptyset_2]_n$. Most electron pairs are localized in the regions between Ge and O, revealing the covalent nature of the Ge–O/Ge– \emptyset bonds. The population of the electron pairs between Ge and O in $[\text{GeO}\emptyset_2]_n$ resembles that in $[\text{Ge}\emptyset_4]_n$ (Fig. R1b), reflecting a similar bonding character in the two structures. The ELF maps around Ge and O are similar in the two maps, which indicates that the redistribution of electrons due to the breakage of the Ge– \emptyset bond in $[\text{Ge}\emptyset_4]_n$ is not significant.

Fig. R1 has been inserted into Fig. 5. The relevant discussion is provided in Lines 126 to 132.

According to the DFT computational results (Figs. 5 and 7 in the Main Text), the π bond is not formed in the $[\text{GeO}\emptyset_2]_n$ chain, probably due to the significant difference in size between the Ge-4p and the O-2p orbitals; therefore, no π electron is localized on oxygen.

Fig. R2 | Hirshfeld charge distributions and valence ELF maps for the $[\text{GeO}\emptyset_2]_n$ and the $[\text{GeO}_2\emptyset_2^{2-}]_n$ chains. a Hirshfeld charge distribution of the $[\text{GeO}\emptyset_2]_n$ chain. **b** Hirshfeld charge distribution of the $[\text{GeO}_2\emptyset_2^{2-}]_n$ chain. **c** ELF of the $[\text{GeO}\emptyset_2]_n$ chain. **d** ELF of the $[\text{GeO}_2\emptyset_2^{2-}]_n$ chain.

The Hirshfeld charge is defined as the difference between the molecular/group and unrelaxed atomic charge densities [Hirshfeld, F. L. Bonded-atom fragments for describing molecular charge densities. *Theoret. Chim. Acta* **44**, 129–138 (1977)]. In the revised version, the Hirshfeld charge, replacing the Mulliken charge, is used to describe the electronic structures of the $[\text{GeO}\emptyset_2]_n$ and the $[\text{GeO}_2\emptyset_2^{2-}]_n$ chains. The results are presented in Figs. R2a and R2b. The strengths of the Ge–O bands in the two chains are evaluated by the valence ELF maps. As shown in Figs. R2c and R2d, the electron pairs around the Ge–O bonds in the $[\text{GeO}\emptyset_2]_n$ chain are less than that in the $[\text{GeO}_2\emptyset_2^{2-}]_n$ chain, indicating that the Ge–O bonds in the $[\text{GeO}\emptyset_2]_n$ chains are weaker than that in the $[\text{GeO}_2\emptyset_2^{2-}]_n$ chain.

Supplementary Fig. 4 has been replaced by Fig. R2 in the Supplementary Information. The relevant discussion is provided in Lines 155 to 159 in the Main Text.

This comment applies for all sections where the electronic structure is discussed within the text.

All of the revisions about the electronic structures have been made and highlighted by yellow color in the revised manuscript.

REVIEWERS' COMMENTS (fourth round)

Reviewer #1 (Remarks to the Author):

I read the revised version of "Threefold coordinated germanium in a GeO₂ melt" by Singing Wan et al.

The authors included in the analysis of the electronic structure a couple of descriptors grounded in physics, i.e. the Electron Localization Function and Hirshfeld charges. Their conclusions on the bonding nature of GeO₂ melt are now given a solid basis, which is a remarkable improvement in the text.

Overall, the quality of the article is now appropriate, and I do believe that this investigation is likely to reach the interest of a wide audience of researchers. I therefore recommend publication of the article without further changes.

Response to the comments from Reviewers (fourth round)

We sincerely thank the reviewer for taking the time and for his/her previous comments which are valuable in improving the quality of our manuscript.

Reviewer #1 (Remarks to the Author):

I read the revised version of "Threefold coordinated germanium in a GeO₂ melt" by Singing Wan et al.

The authors included in the analysis of the electronic structure a couple of descriptors grounded in physics, i.e. the Electron Localization Function and Hirshfeld charges. Their conclusions on the bonding nature of GeO₂ melt are now given a solid basis, which is a remarkable improvement in the text.

Overall, the quality of the article is now appropriate, and I do believe that this investigation is likely to reach the interest of a wide audience of researchers. I therefore recommend publication of the article without further changes.

We are very grateful for the positive assessment and the recommendation for publication in Nature Communications.